# Antimicrobial and Synergistic Effects of Commercial Piperine and Piperlongumine in Combination with Conventional Antimicrobials

**DOI:** 10.3390/antibiotics8020055

**Published:** 2019-05-04

**Authors:** Eunice Ego Mgbeahuruike, Milla Stålnacke, Heikki Vuorela, Yvonne Holm

**Affiliations:** 1Division of Pharmaceutical Biosciences, Faculty of Pharmacy, University of Helsinki, P.O. Box 56, FI-00014 Helsinki, Finland; Heikki.vuorela@helsinki.fi (H.V.); Yvonne.holm@helsinki.fi (Y.H.); 2Department of Pharmacology, Institute of Neuroscience and Physiology, The Sahlgrenska Academy, University of Gothenburg, Box 431, SE-40530 Gothenburg, Sweden; gusstami@student.gu.se

**Keywords:** piperine, piperlongumine, antibacterial, antifungal, synergy

## Abstract

Microbial resistance to currently available antibiotics is a public health problem in the fight against infectious diseases. Most antibiotics are characterized by numerous side effects that may be harmful to normal body cells. To improve the efficacy of these antibiotics and to find an alternative way to minimize the adverse effects associated with most conventional antibiotics, piperine and piperlongumine were screened in combination with conventional rifampicin, tetracycline, and itraconazole to evaluate their synergistic, additive, or antagonistic interactions against *Staphylococcus aureus*, *Pseudomonas aeruginosa,* and *Candida albicans*. The fractional inhibitory concentration index was used to estimate the synergistic effects of various combination ratios of the piperamides and antibiotics against the bacterial and fungal strains. Both piperine and piperlongumine showed synergistic effects against *S. aureus* when combined at various ratios with rifampicin. Synergistic interaction was also observed with piperine in combination with tetracycline against *S. aureus*, while antagonistic interaction was recorded for piperlongumine and tetracycline against *S. aureus.* All the piperamide/antibacterial combinations tested against *P. aeruginosa* showed antagonistic effects, with the exception of piperine and rifampicin, which recorded synergistic interaction at a ratio of 9:1 rifampicin/piperine. No synergistic interaction was observed when the commercial compounds were combined with itraconazole and tested against *C. albicans*. The results showed that piperine and piperlongumine are capable of improving the effectiveness of rifampicin and tetracycline. Dosage combinations of these bioactive compounds with the antibiotics used may be a better option for the treatment of bacterial infections that aims to minimize the adverse effects associated with the use of these conventional antibacterial drugs.

## 1. Introduction

Infectious diseases caused by bacteria and fungi are one of the leading causes of death worldwide [1,2,3]. The emergence of multidrug resistance in microbes and the nonavailability of antibiotics to combat and treat microbial infections have led to a constant search for new antimicrobial agents from natural sources. Previous research has shown that about 13 million deaths are recorded throughout the world each year as a result of bacterial and fungal diseases that are often caused by multidrug-resistant pathogens [4,5]. Natural products of plant origin include novel therapeutic and highly effective antimicrobial agents [6,7]. In clinical practice, one of the leading new advances in the fight against microbial resistance is antimicrobial combination therapy [8,9,10]. Alternatively, bioactive compounds from natural sources could also be screened for leads to the discovery of new antibiotics to combat microbial resistance [5,11,12]. The mode of activity of antimicrobial combination therapy in antimicrobial treatments differs from that of the same individual antibiotics when administered as single drugs [8,13]. Bioactive compounds and extracts from plants can also potentiate the effect of antimicrobial drugs, thereby acting as antibiotic adjuvants [5,11]. Another strategic approach to the fight against multidrug-resistant bacteria in clinical practice is to deactivate the resistance mechanism of the bacteria to the existing antibiotics. In addition, pharmacological studies have shown that synergistic interaction is necessary and could greatly enhance the activity of antimicrobial compounds with moderate or weak activity. Thus, these bioactive plant compounds can serve as inhibitors against multidrug-resistant pathogens [5,14].

Piperine (1) and piperlongumine (2) are novel alkaloids found in most *Piper* species. Piperlongumine (2), also known as piplartin, is a piperamide compound found in Indian long pepper *Piper longum* L. [15]. It is a bioactive compound of clinical importance that is active against multidrug-resistant pathogens such as *Pseudomonas aeruginosa* (Schroeter) Mogul and *Staphylococcus aureus* Rosenbach [16]. Piperine (1) is a piperamide compound found in *Piper guineense* Schumach. & Thonn., black pepper *Piper nigrum* L., and other *Piper* species [17,18,19]. The chemical structures of piperine and piperlongumine are shown in Figure 1. Previous research has shown that piperine is a potent antibacterial agent acting as an inhibitor of the efflux pump of *S. aureus* [20,21]. Recent pharmacological research on piperine and piperlongumine [22,23,24,25] led us to hypothesize that these piperamide compounds may potentiate the antibiotic effect of some conventional antimicrobials, thereby acting as antibiotic adjuvants.

The aim of this study was to investigate the in vitro antimicrobial effects of piperine and piperlongumine when used in combination at various ratios with some conventional antimicrobials (tetracycline, rifampicin, and itraconazole) against *P. aeruginosa, S. aureus*, and *Candida albicans*. It was aimed at evaluating whether piperine and piperlongumine interact synergistically when combined with tetracycline, rifampicin, or itraconazole at various ratios. This is the first report on the synergistic activity of piperlongumine in combination with conventional antimicrobials. The study of the synergistic interaction of these piperamide compounds with conventional antibiotics may aid in addressing the problem of microbial resistance and provide new strategies for improving the use of antibiotics in the treatment of infectious diseases. Piperine and piperlongumine as antibiotic adjuvants may help to reduce the adverse effects associated with these conventional antibiotics. 

## 2. Results

To evaluate an alternative way to fight multidrug-resistant pathogens and to minimize the adverse effects associated with most conventional antibiotics, piperine and piperlongumine were screened singly and in combination with conventional antimicrobials to determine their synergistic, additive, or antagonistic interactions against *S. aureus*, *P. aeruginosa,* and *C. albicans* (Table 1, Figure 2 and Figure 3). The aim was to investigate the synergistic effects of piperine and piperlongumine when used singly and in combination with conventional antimicrobials against *S. aureus*, *P. aeruginosa,* and *C. albicans. S. aureus*, *P. aeruginosa,* and *C. albicans* are pathogens that are capable of causing systemic infections in humans. As such, 90–95% of *S. aureus* strains are resistant to most conventional antibiotics [8,26]. The fractional inhibitory concentration (FIC) index was used to determine the synergistic interaction between the commercial piperamides and the antimicrobials. The FIC index method is one of the most accurate means for determining synergistic interactions when two inhibitors are studied in various combinations [27,28]. A synergistic effect was observed when the FIC index value of the compound of interest was ≤ 0.5 [26]. Previous research has also shown that the ratio of combination of two inhibitors could influence the degree of their interaction, such that the interactions could vary depending on the ratio by which the two inhibitors are combined [28].

The results of the first preliminary agar diffusion method performed singly on the piperamide compounds and antibacterials against *S. aureus* showed that the inhibition zone diameters of tetracycline, rifampicin, piperine, and piperlongumine were 49.1 mm, 47.2 mm, 15.2 mm, and 12.4 mm, respectively (Table 1). Preliminary agar diffusion screening was also performed on the various piperamide/antimicrobial combinations to ascertain the five best combinations to screen for the minimum inhibitory concentrations (MICs) and FIC index calculations for synergistic or antagonistic evaluations (Figure 2 and Figure 3).

The MICs of the various combinations were evaluated (Table 2) to obtain the values used in calculating the FIC indices.

Rifampicin and tetracycline were combined at various ratios with piperine and piperlongumine and tested against *S. aureus.* For piperlongumine, a synergistic effect was observed at a ratio of 5:5 with rifampicin against this bacterium (Table 3). This result demonstrates that the pharmacological interaction between a bioactive compound and a conventional antimicrobial can be affected by the ratio and concentration at which the two are combined. However, antagonistic interactions were observed between piperlongumine and tetracycline at all the combination ratios tested against this bacterium (Table 3).

## 3. Discussion

### 3.1. Antibacterial Synergistic Effects of Piperine and Piperlongumine against S. aureus

For piperine, the FIC index value of rifampicin and piperine against *S. aureus* in this study showed synergistic interaction at a ratio of 3:7 (Table 3). A synergistic effect was observed between piperine and rifampicin and between piperine and tetracycline, as shown in Table 3. Rifampicin is an antibiotic that is often used in the treatment of systemic bacterial infections in antimicrobial therapy, and this antibiotic is characterized by numerous adverse effects when used consecutively for 10–14 days [29,30]. A formulation containing a fixed-dose combination of rifampicin and piperine (rifampicin 200 mg + piperine 10 mg) has been approved in India to minimize the adverse effects associated with the use of conventional rifampicin in the antimicrobial treatment against *Mycobacterium tuberculosis* [30]. Our results show here that piperine could be a compound leading to the discovery of new antibacterial drugs with minimum adverse effects. We observed that piperine and piperlongumine are capable of improving the effectiveness of rifampicin and tetracycline. We suggest, therefore, that dosages of combinations of these bioactive compounds with antibiotics could be a better option for the treatment of bacterial infections than the conventional antibiotics which often show problems with multidrug-resistant pathogens. Our study demonstrates that a combination of antibacterial (rifampicin and tetracycline) with piperamide compounds (piperine and piperlongumine) could lead to minimal systemic toxicity to normal cells during antimicrobial treatments.

### 3.2. Antibacterial Synergistic Effects of Piperine and Piperlongumine against P. aeruginosa

Preliminary agar diffusion assays performed individually with piperamides and antibiotics against *P. aeruginosa* (Table 1) showed that the inhibition zone diameters of tetracycline, rifampicin, piperine, and piperlongumine were 34.6 mm, 30.3 mm, 14.4 mm, and 11.5 mm, respectively. When the MIC of the various combinations was determined, the FIC index calculations showed mainly antagonistic interactions between the piperamides and the antibiotics against *P. aeruginosa*, with the exception of the 9:1 ratio of the piperine/rifampicin combination, which showed synergistic activity (Table 3). *P. aeruginosa* is a Gram-negative pathogenic bacterium that causes systemic infections in immunocompromised individuals and is often associated with antibacterial resistance [31]. For piperlongumine, no synergistic effect was observed in any of the antibiotic combinations (rifampicin and tetracycline) against *P. aeruginosa*, although additive and antagonistic interactions were observed. The results of tetracycline/piperlongumine and rifampicin/ piperlongumine combinations tested at various ratios demonstrated no synergistic effects between tetracycline and piperlongumine as well as between rifampicin and piperlongumine against *P. aeruginosa*. These results suggest that piperine and piperlongumine may not improve the effectiveness of tetracycline against *P. aeruginosa* infections. Indeed, appropriate measures should be taken to avoid too heavy a consumption of herbal drugs or foods rich in piperine and piperlongumine when undergoing antimicrobial treatment with tetracycline against diseases caused by *P. aeruginosa*. 

For piperine, a synergistic interaction with rifampicin was observed for a rifampicin/piperine combination at a ratio of 9:1 against *P. aeruginosa*, while antagonistic or additive interactions were observed for the other combinations tested. Piperine was previously tested for its immunomodulatory activity in enhancing the efficacy of rifampicin in a murine model of *M. tuberculosis* infection, thereby acting in synergy with rifampicin to improve its therapeutic efficacy [32]. However, the synergistic efficacy recorded in our study for a rifampicin/piperine combination showed that the efficacy could be influenced by the ratio of the combination, i.e, the synergistic, additive, or antagonistic interactions were greatly dependent on the ratio between rifampicin and piperine. 

### 3.3. Antifungal Synergistic Effects of Piperine and Piperlongumine against C. albicans

When preliminary agar diffusion assays were performed singly on the piperamides and itraconazole against *C. albicans*, the inhibition zone diameters of piperine, piperlongumine, and itraconazole were 16.5 mm, 18.2 mm, and 15.2 mm, respectively (Table 1). The results of the FIC index evaluations of the various combinations showed additive or antagonistic interactions (Table 3). No synergistic effects were recorded in our study against *C. albicans*. *C. albicans* is an opportunistic pathogen that is responsible for most yeast infections in humans [33]. These results demonstrate that piperine and piperlongumine cannot improve the effectiveness of itraconazole, and thus herbal formulations made from these bioactive compounds should be avoided while undergoing treatments for yeast infections.

## 4. Materials and Methods 

### 4.1. Sources of Antibiotics and Commercial Compounds

Analytical-grade commercial piperlongumine (≥ 97.0% purity) and piperine (≥ 97.0% purity) were purchased from TCI Europe N.V. (Zwijndrecht, Belgium). Tetracycline hydrochloride (Sigma-Aldrich, St. Louis MO, USA), rifampicin (Sigma-Aldrich), and itraconazole (Sigma-Aldrich) were the antibiotics used in the investigation. The selection of tetracycline, rifampicin, and itraconazole was based on their activity against bacterial and fungal strains in a previous study [34]. *S. aureus* American Type Culture Collection ATCC 25923, *P. aeruginosa* ATCC 27853, and *C. albicans* ATCC 10231 cultures used in this investigation were obtained from the Division of Pharmaceutical Biosciences, Faculty of Pharmacy, University of Helsinki, Finland. The bacterial strains were selected because they are significant human pathogens capable of causing life-threatening nosocomial infections [35,36], while *C. albicans*, on the other hand, causes fungal diseases in humans [37].

### 4.2. Preparation of Piperine, Piperlongumine, and Antimicrobials 

Each of the commercial piperamides (piperine and piperlongumine) was prepared to a final concentration of 5 mg/mL (stock solution) in 70% ethanol and tested in the initial screening. An initial stock solution of 5 mg/mL was prepared for each of the antibiotics (tetracycline, rifampicin, and itraconazole). Piperine and piperlongumine were also prepared in combination with tetracycline, rifampicin, or itraconazole at various ratios according to the method of Van Vuuren et al. [10]. The combinations/dilutions were prepared at nine different ratios of 1:9, 2:8, 3:7, 4:6, 5:5, 6:4, 7:3, 8:2, 9:1 as follows: rifampicin/piperlongumine, rifampicin/piperine, tetracycline/piperlongumine, tetracycline/piperine, itraconazole/piperine, and itraconazole/piperlongumine. 

### 4.3. Agar Disk-Diffusion Method

The agar disk-diffusion method was applied for the initial screening of the nine combination ratios according to the method of the Clinical and Laboratory Standards Institute [38]. Sterile Petri dishes (∅ = 14 cm, VWR International Oy, Helsinki, Finland) were used for the screening. For the antibacterial screening against *S. aureus and P. aeruginosa*, 25 mL of sterile base agar (Antibiotic agar No. 2, Difco, VWR) was applied as a bottom layer into the sterile Petri dishes, using a sterile serological pipet (Falcon; BDLabware Europe) and allowed to solidify. Thereafter, 25 mL of isosensitest agar (OXOID, ThermoFisher Scientific, Waltham, MA, USA) was applied as the top layer. For the antifungal screening, 25 mL of sterile base agar (Antibiotic agar No. 2, Difco VWR) was applied as a bottom layer into the sterile Petri dishes and 25 mL of Saboraud agar (OXOID, ThermoFisher Scientific, Basingstoke, United Kingdom) was applied as the top layer. The agar in the Petri dishes was allowed to solidify and then stored at +4 °C. The screening was initiated with the inoculation of the bacterial strains or *C. albicans* onto solid nutrient or Saboraud agar slants that were incubated overnight for the bacterial strains and 48 h for the fungus at +37 °C. Viable bacterial and fungal cultures from the agar slants were used to prepare an inoculum for the test. Bacterial and fungal specimens from the agar slants were transferred into 2 mL of a 0.9% (*w/v*) sodium chloride (NaCl) solution in a sterile glass tube, using a sterile inoculation loop. In all, 1 mL of the suspension was transferred into another sterile glass tube, and the absorbance was measured at 625 nm (UV–Visible Spectrophotometer, Pharmacia LKB-Biochrom 4060; Pfizer Inc., New York, NY, USA). The other milliliter of the suspension (sterile part) was diluted with the 0.9% NaCl solution so that the absorbance at 625 nm became 0.1 (this suspension contained approximately 1.5 × 10^8^ colony-forming units (CFUs)/mL). In all, 200 µL of this diluted bacterial or fungal suspension were spread evenly on each Petri dish and left to dry for several seconds with the lid open. A sterile cork borer (11 mm diameter) was used to make six equidistant holes on the agar surface of the Petri dishes. A total of 200 µL of the 5 mg/mL concentration of each of the piperamides and 200 µL of the 10 mg/mL antimicrobial were carefully pipetted into the holes; 70% ethanol was used as the solvent control. The first screening of the combined samples was done in the same manner as described above for the agar diffusion method, and the various inhibition zones were recorded. The Petri dishes were transferred to the cold room and incubated at +4 °C for 1 h. Thereafter, they were incubated at +37 °C, overnight for the bacterial strains and 48 h for the fungal strain. The diameters of the zones of inhibition were measured with a caliper under a Petri dish magnifier and expressed as the mean of the diameters of three replicates ± the standard error of the mean (SEM). 

### 4.4. Microdilution Method for Minimum Inhibitory Concentration Estimation 

The MIC was estimated for the five best combinations of each piperamide/antimicrobial. The MIC was taken as the lowest concentration of the piperamide/antimicrobial resulting in the inhibition of at least 90% of the growth of the bacterial or fungal strain. The MIC values were determined using a microdilution turbidimetric broth method based on the guidelines of the Clinical and Laboratory Standards Institute [38]. For MIC evaluation, two-fold serial dilutions from 2500 to 9.75 µg/mL were prepared in sterile Mueller–Hinton broth (bacterial strains) and Saboraud broth (*C. albicans*) for each of the piperamide/antimicrobial combinations. The commercially pure compounds, piperine and piperlongumine (5 mg/mL concentrations in 70% ethanol), were two-fold serially diluted in Mueller–Hinton broth for the bacterial strains and Saboraud broth for the fungal strain. Tetracycline, rifampicin, and itraconazole were similarly diluted in Mueller–Hinton broth or Saboraud broth from 125 to 0.48 µg/mL, respectively; 96-well microtiter plates (Nunc, Nunclone; Nalge Nunc International, Roskilde, Denmark) were used for the tests. The bacterial or fungal cultures were inoculated in 5 mL of Mueller–Hinton broth and grown for 24 h at +37 °C for the bacterial strains or in 5 mL of Saboraud broth and grown for 48 h at +37 °C for *C. albicans* before the test. Absorbance of 1 mL of the overnight bacterial culture was measured for turbidity at 625 nm, using a UV–Visible Spectrophotometer type 1510 (ThermoFisher Scientific). The absorbance was adjusted to 0.1 at 625 nm (approximately 1.0 × 10^8^ CFUs/mL). A total of 100 µL of this suspension A_625_ = 0.1 was further diluted a hundredfold to obtain a working suspension or inoculum containing 1.0 × 10^6^ CFUs/mL. In all, 100 µL of this inoculum and 100 µL of each of thepiperamide/antimicrobial combinations and solvent controls were introduced into the 96-well microtiter plates. Each well therefore contained 5 × 10^5^ CFUs/mL. The growth control (GC) wells contained only the bacterial suspension, and the test (T) wells contained the piperamide/antimicrobial + bacterial suspension. Moreover, negative control wells were prepared for each piperamide/antimicrobial combination ratio tested, and these wells contained the piperamide/antimicrobial combinations at different ratios and the broth only. The microwell plates were incubated for 24 h in an incubator coupled to a shaker at +37 °C. The turbidity of the wells at 620 nm was recorded using a Victor 1420 spectrophotometer (PerkinElmer (Wallac Oy), Turku, Finland). The tests were done in triplicate, and the percentage growth was expressed as the mean of these triplicates ± the SEM. 

### 4.5. Fractional Inhibitory Concentration Index 

The FIC index was used to estimate the synergy [5,26]. For the FIC index calculation, the MIC values of five combinations of each piperamide/antimicrobial at different ratios were used to calculate FIC (A) and FIC (B), using the equation below:

The FIC values were calculated as follows:
$$\begin{array}{c} FIC\left(A\right)=\frac{MICvalueofcombinedpiperamideandantimicrobial}{MICvalueofantimicrobialalone}\\ FIC\left(B\right)=\frac{MICvalueofcombinedpiperamideandantimicrobial}{MICvalueofpiperamidealone}\\ FICindex=FIC\left(A\right)+FIC\left(B\right) \end{array}$$

A FIC index value ≤ 0.5 was interpreted as synergy. The FIC index was calculated on the basis of the above-mentioned equation, in which the FIC index = X + Y and the interactions are defined as: FIC index ≤ 0.5, synergy; > 0.5 to ≤ 1.0, additive; > 1.0 to ≤ 2.0, indifferent; and > 2.0, antagonistic [26].

## 5. Conclusions

Piperine and piperlongumine are possible antibiotic adjuvants that should be applied in drug discovery for use as effective antibacterial treatments. The synergistic efficacy of piperlongumine and piperine in combination with rifampicin has shown that these two piperamides are bioactive scaffolds that could aid in fighting against multidrug-resistant pathogens, thereby improving the effectiveness of rifampicin in the treatment of infectious diseases. Our results show that the synergistic efficacy of piperamide compounds is dependent on the concentration and ratio of the combination. This indicates that the use of piperine and piperlongumine may minimize the adverse effects associated with conventional tetracycline and rifampicin and improve the effectiveness of these antibiotics. However, the antagonistic effects observed with the commercial compounds/antibiotics used against *P. aeruginosa* in this study demonstrate that appropriate caution should be taken to avoid the consumption of herbal drugs made from *Piper* species during antibacterial treatment therapies against *P. aeruginosa* infections. We suggest that piperine and piperlongumine could be considered as therapeutic bioactive compounds for antibacterial drug discovery. Further studies should be conducted on the mode of activity of the synergistic interaction between piperamides and antibiotics.

## Figures and Tables

**Figure 1 antibiotics-08-00055-f001:**
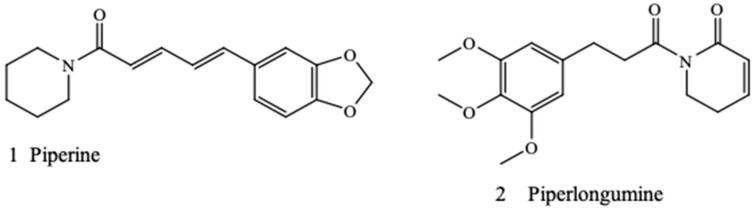
Chemical structures of piperine and piperlongumine.

**Figure 2 antibiotics-08-00055-f002:**
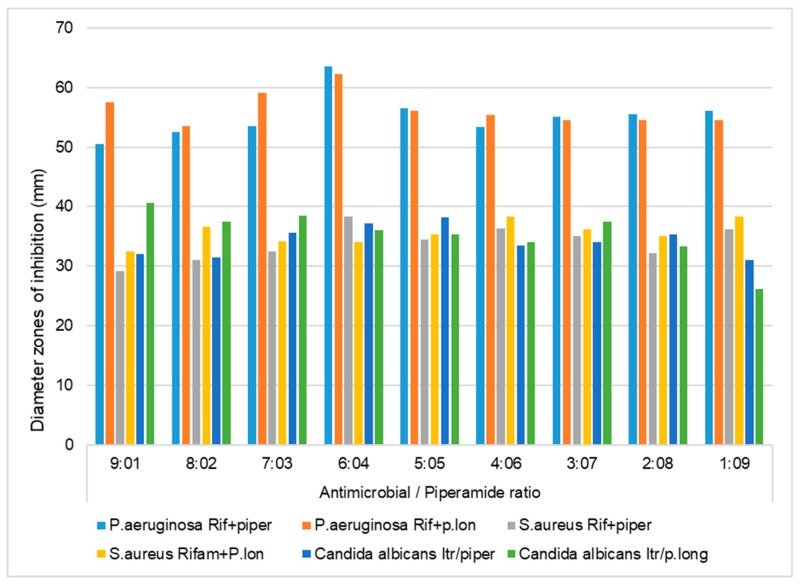
Inhibition zone diameters (mm) of the antibacterial and antifungal effects of piperine and piperlongumine combined at various ratios with rifampicin/itraconazole against *S. aureus*, *P. aeruginosa*, and *C. albicans*, using the agar diffusion method. Rif, rifampicin; Itr, itraconazole; piper, piperine; p.long, piperlongumine; 200 µL of the piperamide/antimicrobial combinations were applied in the wells.

**Figure 3 antibiotics-08-00055-f003:**
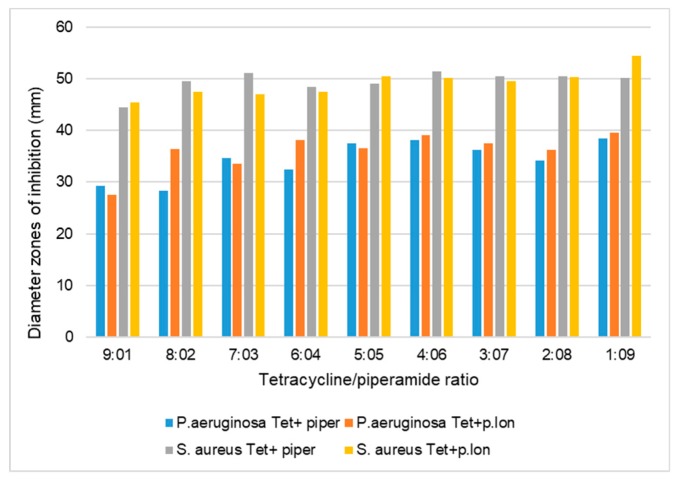
Inhibition zone diameters (mm) of the antibacterial effects of piperine and piperlongumine combined at various ratios with tetracycline against *S. aureus* and *P. aeruginosa*, using the agar diffusion method. The inhibition zone diameters (mm) are the means of three replicates (*n* = 3). P.lon, piperlongumine; tet, tetracycline; piper, piperine.

**Table 1 antibiotics-08-00055-t001:** Inhibition zone diameters (mm) of the antibacterial and antifungal effects of piperine, piperlongumine (P. longumine), rifampicin, tetracycline, itraconazole, and 70% ethanol tested individually against *Staphylococcus aureus*, *Pseudomonas aeruginosa*, and *Candida albicans*, using the agar diffusion method.

Bacteria/ Fungi	Piperine	P. longumine	Tetracycline	Rifampicin	70% Ethanol	Itraconazole
*S. aureus*	15.2 ± 0.31	12.4 ± 0.31	49.1 ± 0.32	47.2 ± 0.17	NA	NP *
*P. aeruginosa*	14.4 ± 0.33	11.5 ± 0.33	34.6 ± 0.33	30.3 ± 0.33	NA	NP *
*C. albicans*	16.5 ± 0.31	18.2 ± 0.31	NP *	NP *	NA	15.2 ± 0.33

NA, not active; *, not performed. The inhibition zone diameters (mm) represent the means of three replicates (*n* = 3) ± the SEM (standard error of the mean).

**Table 2 antibiotics-08-00055-t002:** Minimum inhibitory concentrations (MIC, µg/mL) of piperine and piperlongumine combined at various ratios with rifampicin (rifam), tetracycline (tetracy), itraconazole (itracon) against *S. aureus*, *P. aeruginosa*, and *C. albicans*.

Ratioof Antibiotics/Compound	*S. aureus*	*P. aeruginosa*	*C. albicans*
Rifam + piperine 1:9	7.8	15.6	NP*
Rifam + piperine 3:7	1.9	15.6	NP *
Rifam + piperine 5:5	3.9	62.5	NP *
Rifam + Piperine 7:3	1.9	62.5	NP *
Rifam + piperine 9:1	7.8	15.6	NP *
Rifam + piperlongumine 1:9	7.8	31.2	NP *
Rifam + piperlongumine 3:7	125	15.6	NP *
Rifam + piperlongumine 5:5	3.9	15.6	NP *
Rifam + piperlongumine 7:3	62.5	15.6	NP *
Rifam + piperlongumine 9:1	62.5	7.8	NP *
Tetracy + piperine 1:9	7.8	62.5	NP *
Tetracy + piperine 3:7	3.9	31.2	NP *
Tetracy + piperine 5:5	0.97	15.6	NP *
Tetracy + piperine 7:3	1.9	7.8	NP *
Tetracy + piperine 9:1	7.8	7.8	NP *
Tetracy + piperlongumine 1:9	31.2	31.2	NP *
Tetracy + piperlongumine 3:7	3.9	7.8	NP *
Tetracy + piperlongumine 5:5	3.9	15.6	NP *
Tetracy + piperlongumine 7:3	1.9	7.8	NP *
Tetracy + piperlongumine 9:1	62.5	7.8	NP *
Itracon + piperine 1:9	NP *	NP *	3.9
Itracon + piperine 3:7	NP *	NP *	3.9
Itracon + piperine 5:5	NP *	NP *	7.8
Itracon + piperine 7:3	NP *	NP *	7.8
Itracon + piperine 9:1	NP *	NP *	15.6
Itracon + piperlongumine 1:9	NP *	NP *	31.2
Itracon + piperlongumine 3:7	NP *	NP *	7.8
Itracon + piperlongumine 5:5	NP *	NP *	7.8
Itracon + piperlongumine 7:3	NP *	NP *	3.9
Itracon + piperlongumine 9:1	NP *	NP *	31.2
Piperine only	3.9	15.6	7.8
Piperlongumine only	15.6	31.2	3.9
Rifampicin only	1.97	0.48	NP *
Tetracycline only	0.97	0.97	NP *
Itraconazole only	NP *	NP *	15.6

The MIC values represent the means of triplicates. Only five combinations were tested for MIC. *, not performed.

**Table 3 antibiotics-08-00055-t003:** Antimicrobial synergistic activity of piperine and piperlongumine against *S.aureus*, *P. aeruginosa,* and *C. albicans*.

Ratio	FIC Index		
	*S. aureus*	*P. aeruginosa*	*C. albicans*
Rifam + piperine 1:9	5.9 (AT)	33.5 (AT)	NP *
Rifam + piperine 3:7	0.2 (S)	33.5 (AT)	NP *
Rifam + piperine 5:5	3.9 (AT)	8.0 (AT)	NP *
Rifam + piperine 7:3	1.2 (I)	5.0 (AT)	NP *
Rifam + piperine 9:1	12.6 (AT)	0.4 (S)	NP *
Rifam + piperlongumine 1:9	4.4 (AT)	66.0 (AT)	NP *
Rifam + piperlongumine 3:7	39.0 (AT)	33.0 (AT)	NP *
Rifam + piperlongumine 5:5	0,5 (S)	1.5 (I)	NP *
Rifam + piperlongumine 7:3	17.0 (AT)	2.0 (I)	NP *
Rifam + piperlongumine 9:1	17.0 (AT)	1.0 (I)	NP *
Tetracy + piperine 1:9	10.0 (AT)	68.0 (AT)	NP *
Tetracy + piperine 3:7	4.5 (AT)	32.0 (AT)	NP *
Tetracy + piperine 5:5	0.3 (S)	0.7 (A)	NP *
Tetracy + piperine 7:3	2.4 (AT)	0.7 (A)	NP *
Tetracy + piperine 9:1	12.1 (AT)	1.5 (I)	NP *
Tetracy + piperlongumine 1:9	34.1 (AT)	33.1 (AT)	NP *
Tetracy + piperlongumine 3:7	4.1 (AT)	8.2 (AT)	NP *
Tetracy + piperlongumine 5:5	1.1 (I)	2.5 (AT)	NP *
Tetracy + piperlongumine 7:3	0.9 (A)	1.5 (I)	NP *
Tetracy + piperlongumine 9:1	48.9 (AT)	1.5 (I)	NP *
Itracon + piperine 1:9	NP *	NP *	0.8 (A)
Itracon + piperine 3:7	NP *	NP *	1.2 (I)
Itracon + piperine 5:5	NP *	NP *	4.0 (AT)
Itracon + piperine 7:3	NP *	NP *	3.0 (AT)
Itracon + piperine 9:1	NP *	NP *	4.0 (AT)
Itracon + piperlongumine 1:9	NP *	NP *	10.1 (AT)
Itracon + piperlongumine 3:7	NP *	NP *	0.7 (A)
Itracon + piperlongumine 5:5	NP *	NP *	1.3 (I)
Itracon + piperlongumine 7:3	NP *	NP *	1.0 (I)
Itracon + piperlongumine 9:1	NP *	NP *	12.0 (AT)

The activity is expressed as the fractional inhibition concentration (FIC) index, which is calculated from the MIC of the various piperamide/antimicrobial combinations. The FIC index interactions were defined as: FICI ≤ 0.5, (S) synergy; >0.5 to ≤1.0, (A) additive; >1.0 to ≤2.0, (I) indifferent; and >2.0, (AT) antagonistic [26]. *, not performed.

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
