# Peer review of "Antimicrobial and Synergistic Effects of Commercial Piperine and Piperlongumine in Combination with Conventional Antimicrobials"

_antibiotics, 2019, doi:10.3390/antibiotics8020055_

Round 1

Reviewer 1 Report

This is an interesting and well written manuscript that describes the effect piperine and piperlongumine in combination with conventional antimicrobials.

However, in my opinion, the authors should explain in more detail the following questions

-In Results section

In its current form, tables 2 and 3 are unclear. Maybe the use of graphics instead of tables could clarify the results

-In Material and Methods section

In Agar disk-diffusion method, what was the initial concentration of the antibiotics??

*In Discussion section

-I believe the discussion should be expanded with the studies of other authors

Please review the following articles in this regard

1.       Freitas CLA, Santos FFP, Dantas-Junior OM, Inácio VV, Matias EFF,

Quintans-Júnior LJ, Aguiar JJS, Coutinho HDM. Enhancement of antibiotic activity

byphytocompounds of Turnerasubulata. Nat Prod Res. 2019 Feb 14:1-5.

2.       Coutinho HD, Costa JG, Lima EO, Falcão-Silva VS, Siqueira JP Jr. Herbal

therapy associated with antibiotic therapy: potentiation of the antibiotic

activity against methicillin--resistant Staphylococcus aureus by Turnera

ulmifolia L. BMC Complement Altern Med. 2009 May 8;9:13.

3.       Coutinho HD, Costa JG, Lima EO, Falcão-Silva VS, Siqueira-Júnior JP.

Potentiatingeffect of Menthaarvensis and chlorpromazine in theresistanceto

aminoglycosides of methicillin-resistantStaphylococcusaureus. In Vivo. 2009

Mar-Apr;23(2):287-9. PubMed PMID: 19414416.

4.       Zuo GY, Wang CJ, Han J, Li YQ, Wang GC. Synergism of coumarins from the

Chinese drug Zanthoxylumnitidum with antibacterial agents against

methicillin-resistant Staphylococcus aureus (MRSA). Phytomedicine. 2016 Dec

15;23(14):1814-1820.

Author Response

Reviewer #1

Comment

This is an interesting and well written   manuscript that describes the effect piperine and piperlongumine in   combination with conventional antimicrobials.

However, in my opinion, the authors should   explain in more detail the following questions

-In Results section

In its current form, tables 2 and 3 are   unclear. Maybe the use of graphics instead of tables could clarify the   results

Answer

The   authors have now used graphics to clarify the results in Table 2 and Table 3.   Table 2 and Table 3 have been changed to figure 2 and figure 3. See figure 2   and 3 in the revised manuscript page 4 and page 5.

Comment

In Material and Methods section

In Agar disk-diffusion method, what was the   initial concentration of the antibiotics??

Answer

The   initial concentration of the antibiotics in Agar disk-diffusion method is 5   mg/mL. This is included in the materials and method section page 10 of the   revised manuscript.

Comment

*In Discussion section

-I believe the discussion should be expanded with   the studies of other authors

Please review the following articles in this   regard

1.       Freitas   CLA, Santos FFP, Dantas-Junior OM, Inácio VV, Matias EFF, Quintans-Júnior LJ,   Aguiar JJS, Coutinho HDM. Enhancement of antibiotic activity by phytocompounds   of Turnerasubulata. Nat Prod Res. 2019 Feb 14:1-5.

2.         Coutinho HD, Costa JG, Lima EO, Falcão-Silva VS, Siqueira JP Jr. Herbal therapy   associated with antibiotic therapy: potentiation of the antibiotic activity   against methicillin--resistant Staphylococcus aureus by Turnera ulmifolia L.   BMC Complement Altern Med. 2009 May 8;9:13.

3.         Coutinho HD, Costa JG, Lima EO, Falcão-Silva VS, Siqueira-Júnior JP. Potentiatingeffect   of Menthaarvensis and chlorpromazine in theresistanceto aminoglycosides of   methicillin-resistantStaphylococcusaureus. In Vivo. 2009 Mar-Apr;23(2):287-9.   PubMed PMID: 19414416.

4.       Zuo GY,   Wang CJ, Han J, Li YQ, Wang GC. Synergism of coumarins from the Chinese drug   Zanthoxylumnitidum with antibacterial agents against methicillin-resistant   Staphylococcus aureus (MRSA). Phytomedicine. 2016 Dec 15;23(14):1814-1820.

Answer

The   authors have reviewed these articles and are in agreement with their   findings.

Reviewer 2 Report

This manuscript describes the antimicrobial and synergistic effects of commercial piperine and piperlongumine in combination with conventional antibiotics against a wide range of pathogens. The manuscript is well written, but I think it needs certain improvement before publication

The authors should provide MIC data for each compound tested in the work. Only providing MIC and FICI is not sufficient

I do not agree with some conclusions drawn by the authors. For example, the authors claimed that most combination therapy resulted in antagonistic effect against P. aeruginosa, but not against S. aureus. However, there are 14 antagonistic effects in the case of S. aureus, and only 11 antagonistic effects in the case of P.aeruginosa shown in Table 5. In addition, the authors only considered synergistic effect was beneficial. In fact, additive effect is also an advantageous result for the combination therapy. Therefore, I think the authors should rework on their discussion of the data presented in Table 5.

Author Response

Reviewer #2

Comment

This manuscript describes the antimicrobial   and synergistic effects of commercial piperine and piperlongumine in   combination with conventional antibiotics against a wide range of pathogens.   The manuscript is well written, but I think it needs certain improvement   before publication

The authors should provide MIC data for each   compound tested in the work. Only providing MIC and FICI is not sufficient

Answer

The   authors provided the MIC data of the compounds tested in the work and also   the MIC of the various combinations of the antimicrobial/piperamide, See the   end of Table 2 in page 5 of the revised manuscript.

Comment

I do not agree with some conclusions drawn by   the authors. For example, the authors claimed that most combination therapy   resulted in antagonistic effect against P. aeruginosa, but not against S.   aureus. However, there are 14 antagonistic effects in the case of S. aureus,   and only 11 antagonistic effects in the case of P.aeruginosa shown in Table   5. In addition, the authors only considered synergistic effect was   beneficial. In fact, additive effect is also an advantageous result for the   combination therapy. Therefore, I think the authors should rework on their   discussion of the data presented in Table 5.

Answer

The   authors have added some comments in the discussion section with regards to   this observation.

Reviewer 3 Report

The authors performed several tests and wrote it well. However, minor revision is suggested.

What is NP (not performed??) in the tables. It should be specified using star (*) below the all tables. 

Some graphs instead of tables should be added to see some visual comparison between the samples since only using table will be exhausting for readers.

Author Response

Reviewer #3

Comment

The authors performed several tests and wrote   it well. However, minor revision is suggested.

What is NP (not performed??) in the tables. It   should be specified using star (*) below the all tables. 

Answer

The   authors have specified the NP (not performed??) with star   (*) in the tables and below the tables, See Table 2 and Table 3 in page 5 and   page 7 of the revised manuscript.

Comment

Some graphs instead of tables should be added   to see some visual comparison between the samples since only using table will   be exhausting for readers.

Answer

The   authors have used graphics to show visual comparism between the samples,   Table 2 and Table 3 have been changed to figure 2 and figure 3. See the   revised manuscript page 4 and page 5.

Reviewer 4 Report

The paper demonstrates a nice process where synergism of antimicrobials can be induced with the addition of somewhat benign amide containing small molecules to existing antimicrobials. The article is interesting and is well written and structured, consequently, I would recommend publication following some minor edits highlighted below and in the attached manuscript.

·     Minor amendments of English grammar and sentence structure throughout. 

·     Some minor structural edits including in-text citations of figures and utilising the molecular numbering in-text. Also, table structures which have undergone some minor shifts upon formatting.

Author Response

Reviewer #4

Comment

The paper demonstrates a nice process where   synergism of antimicrobials can be induced with the addition of somewhat   benign amide containing small molecules to existing antimicrobials. The   article is interesting and is well written and structured, consequently, I   would recommend publication following some minor edits highlighted below and   in the attached manuscript.

·     Minor   amendments of English grammar and sentence structure throughout. 

·     Some minor   structural edits including in-text citations of figures and utilising the   molecular numbering in-text. Also, table structures which have undergone some   minor shifts upon formatting.

Answer

The   manuscript   has been grammatically checked for proper language use. The   manuscript has been structurally edited and there is now an in-text citation   of all the tables and figures. Table 2 and 3 have been changed to figure 2   and figure 3 for more visual comparism of the results.  See the corrections highlighted in yellow in   the corrected manuscript.